# Language Development in Preschool Duchenne Muscular Dystrophy Boys

**DOI:** 10.3390/brainsci12091252

**Published:** 2022-09-16

**Authors:** Daniela Pia Rosaria Chieffo, Federica Moriconi, Ludovica Mastrilli, Federica Lino, Claudia Brogna, Giorgia Coratti, Michela Altobelli, Valentina Massaroni, Giulia Norcia, Elisabetta Ferraroli, Simona Lucibello, Marika Pane, Eugenio Mercuri

**Affiliations:** 1Clinical Psychology Unit, Fondazione Policlinico Universitario A. Gemelli IRCCS, 00168 Rome, Italy; 2Psychology, Catholic University of the Sacred Heart, 00168 Rome, Italy; 3Pediatric Neurology, Catholic University, 00168 Rome, Italy; 4Centro Clinico Nemo, Fondazione Policlinico Universitario A. Gemelli IRCCS, 00168 Rome, Italy; 5Department of Safety and Bioethics, Infectious Diseases Institute, Catholic University of Sacred Heart, 00168 Rome, Italy

**Keywords:** Duchenne muscular dystrophy, dystrophin isoforms, visual attention, auditory attention, language development

## Abstract

Background: the present study aims to assess language in preschool-aged Duchenne muscular dystrophy (DMD) boys with normal cognitive quotients, and to establish whether language difficulties are related to attentional aspects or to the involvement of brain dystrophin isoforms. Methods: 20 children aged between 48 and 72 months were assessed with language and attention assessments for preschool children. Nine had a mutation upstream of exon 44, five between 44 and 51, four between 51 and 63, and two after exon 63. A control group comprising 20 age-matched boys with a speech language disorder and normal IQ were also used. Results: lexical and syntactic comprehension and denomination were normal in 90% of the boys with Duchenne, while the articulation and repetition of long words, and sentence repetition frequently showed abnormal results (80%). Abnormal results were also found in tests assessing selective and sustained auditory attention. Language difficulties were less frequent in patients with mutations not involving isoforms Dp140 and Dp71. The profile in Duchenne boys was different form the one observed in SLI with no cognitive impairment. Conclusion: The results of our observational cross-sectional study suggest that early language abilities are frequently abnormal in preschool Duchenne boys and should be assessed regardless of their global neurodevelopmental quotient.

## 1. Introduction

Duchenne muscular dystrophy (DMD) is a hereditary, X-linked, progressive muscular disorder. Over the last few years, increasing attention has been given to the involvement of the central nervous system in DMD and to the role of specific dystrophin isoforms expressed in the brain on neurodevelopment. Several studies explored various aspects of possible brain involvement, including cognitive, behavioral, and psychiatric features [1,2,3,4].

One-third of DMD patients have borderline IQ or intellectual disability. The low IQ cannot be justified by muscle weakness that may potentially affect items involving motor abilities as, even when excluding the motor component, the IQ remains low [5,6], and verbal IQ is generally more impaired than performance IQ. Only a few studies have focused on language, identifying specific difficulties in the phonetic–phonologic area or in receptive language, often found in boys with lower IQ levels [7]. Others explored possible associations with working memory and executive functions that may prevent linguistic processing [8,9].

Most of these studies were performed mainly at school age, when specific language tests are available. Despite the fact that language delay was recognized as one of the early signs of DMD [10,11,12], little attention is paid to earlier aspects of language development [13]. In a retrospective study investigating milestones in DMD boys, first words and full sentences were reported as delayed in 38% and 43%, respectively [14]. Another study reported lower expressive language skills related to impairments of phonological abilities in DMD preschool children compared to healthy siblings [15]. Other information on early aspects of speech and language can be extrapolated by more general studies using neurodevelopmental scales in young DMD boys. Two studies, one using Griffiths developmental scales and the other using Bayley scales, found overall lower developmental quotients compared to typically developing children. In both studies the lower DQ also included lower language skills [16,17,18]. Establishing whether early language disorders can also be present in children with overall normal developmental or cognitive quotients has not been established. 

The aim of this study was to assess language in preschool-age DMD boys with normal global neurodevelopmental or cognitive quotients by using a battery of tests specifically designed to assess language in this age band. We were also interested in establishing (i) whether early language difficulties, if present, differed from those observed in typically developing children of the same age with a language disorder diagnosis and normal IQ; (ii) the possible association with other attentional or behavioral aspects that are frequent in DMD and with the involvement of brain dystrophin isoforms. 

## 2. Materials and Methods

### 2.1. Participants

All patients were affiliated to the child neurology and clinical psychology units of the Policlinico Universitario A. Gemelli Foundation. Patients were recruited and evaluated between 2017 and 2020. Children were included if they had a genetically proven DMD diagnosis, and their age was between 48 and 72 months. As the aim was to establish if language difficulties could also be present in the absence of a more global neurodevelopmental delay, we only included boys with normal IQ/DQ in this study. IQ/DQ was available in all patients as, depending on the age, we routinely perform neurodevelopmental or cognitive scales in all DMD boys. The age of 48 months was determined, as this is the lowest age level for which normative data for the language scale are available. Patients with severe behavioral problems that could interfere with the assessments were excluded. 

In total, 20 DMD children (mean age 5.3 years) fulfilled the inclusion criteria and were included in the study. 

A control group of 20 typically developing boys with a specific language impairment (SLI), such as phonological and expressive language difficulties, but with normal IQ in the same age range as that of our study cohort were used for comparison. 

The cohort of DMD patients was split according to the involvement of dystrophin isoforms that is determined by the site of the genetic mutations in the DMD gene. Patients with mutations upstream of exon 44 have reduced expression of Dp 427 only, mutations after exon 51 affect both Dp427 and Dp140, and mutations after exon 63 have reduced expression of Dp427, Dp140, and Dp71. Mutations between 44 and 51 have recently been reported separately, as the involvement of Dp 140 in these patients is not always clear. 

The study was approved by the institutional ethics committee of our center. Interested patients provided informed consent for the submission of the article elaborated on the obtained data.

### 2.2. Clinical Assessment

The protocol includes a specifically designed battery to assess language for the Italian population of children from the age of 48 months, and other tests assessing attention. Assessment was performed as follows for both the study and the control group.

Language assessment: language abilities were assessed using items from the Batteria per la Valutazione del Linguaggio in Bambini dai 4 ai 12 anni; BVL 4_12 [19]. The test evaluates the two components of production and reception in children aged from 4 to 12 by systematically assessing language skills and the level of linguistic competence in Italian.

From BVL_4–12, we specifically assessed oral comprehension skills (lexical comprehension: comprehension in grammar), oral and language production skills (denomination and articulation), and oral repetition skills (word and sentence repetition).

A further assessment was also performed for speech sound articulation in order to detect possible difficulties in articulating distinct speech sounds. 

Working memory assessment: the nonword repetition test was performed as part of the Batteria per la Valutazione del Linguaggio in Bambini dai 4 ai 12 anni: BVL 4_12 [19]. The digit span test of Batteria per la Valutazione dell’Attenzione Uditiva e della Memoria di Lavoro Fonologica nell’Età Evolutiva, VAUMeLF [19] was also administered.

Attention ability assessment: visual attention was assessed using the Bell test, which is a cancellation task that measures selective attention and visuospatial abilities in children aged from 6 to 14 [20].

Selective and sustained auditory attention was assessed using the auditory continuous performance test (ACPT) and selective attention test (TS1 e TS2) from the Batteria per la Valutazione dell’Attenzione Uditiva e della Memoria di Lavoro Fonologica nell’Età Evolutiva, VAUMeLF [21] for children from 4 to 8 years. The continuous auditory performance test measures the ability to attend to auditory stimuli for an extended period. One such test comprises six trial lists of one-syllable words. The words are presented rapidly, and the children are instructed to speak each time that a target word occurs. The number of incorrect responses for the entire test provides a measure of overall attention ability.

### 2.3. Statistical Analysis

The Mann–Whitney U test was used to compare differences in median scores between the group of children with specific language impairment (SLI) and the group of children with Duchenne muscular dystrophy (DMD).

Depending on the sample size, the chi-squared test or Fisher’s exact test (FET) was used to compare and analyze differences in tasks between SLI and DMD patients, and within the DMD group according to the four clinical variations for specificity with respect to mutation (Duchenne muscular dystrophy upstream of exon 44; Duchenne muscular dystrophy exons 45–51; Duchenne muscular dystrophy exon >51; Duchenne muscular dystrophy exon >62). Task response was subdivided into nonpathological (NP) if results were between the −2nd and 2nd standard deviation (SD), and pathological (P) if above ±2 SD. A two-tailed *p* value < 0.05 was considered to be statistically significant.

## 3. Results

### 3.1. DMD Group

#### 3.1.1. Language Assessment

Table 1 and Figure 1 report the details of the results of the language tests in the studied cohort.

Oral comprehension skills: Abnormal scores (−2SD) on lexical and syntactic comprehension were found in 2 patients (10%). Both had mutations beyond exon 63.

Language production/oral production skills: abnormal scores (−2SD) on denomination were found in 1 patient (5%), who had a mutation between exons 51 and 62. 

Abnormal scores (−2SD) on articulation were found in 16 of the 20 DMD patients (80%), including 5 of the 9 patients with mutations upstream of exon 44, and all the 11 patients with mutations after exon 44.

Speech sound articulation test: 6 of the 20 patients (30%) did not present the voiced vibrant liquid sound (r), 5 of the 20 patients (25%) did not produce unvoiced palatoalveolar fricative phoneme (∫), and 4 of 20 patients (20%) did not present alveolar, labiodental, and alveolar affricate phonemes. 

Oral repetition skills: Abnormal scores (−2SD) on word repetition were found in 4 patients (20%), comprising 2 patients with mutations after exon 51, and 2 patients with mutations beyond exon 63. 

Abnormal scores (−2SD) on sentence repetition were found in 16 of the 20 DMD patients (80%), including 5 of the 9 patients with mutations upstream exon 44, and all the 11 patients with mutations after exon 44 (Table 1). 

#### 3.1.2. Working Memory and Visual-Motor Integration Ability

Abnormal scores (−2SD) were found in 40% of the DMD patients when assessed on nonword repetition (Table 1).

The abnormal scores on word repetition were more frequently found in patients with mutations after exon 44 (Table 1).

#### 3.1.3. Attention Skills

Table 1 and Figure 2 reports details of the results of the attention tests in the studied cohort.

Abnormal scores on auditory attention were found in 16 patients (80%), including 7 of the 9 with mutations upstream exon 44, and all 11 patients with mutations after exon 44. 

Abnormal scores on visual attention were found in 14 patients (70%) including 5 of the 9 with mutations upstream exon 44, 3 of the 5 with mutations between exon 44, and in all 6 with mutations after exon 51. 

### 3.2. LSD Group

Table 1, and Figure 1 and Figure 2 show the results of the tests administered in the control SLI group.

### 3.3. Comparison of the Tests Administered between the DMD Group and the SLI Group

The difference in median score between the DMD and SLI groups was significant for the following administered tests: word repetition (*p* = 0.017), nonword repetition (*p* = 0.002), visual attention (rapidity) (*p* < 0.001), the second auditory attention task (*p* = 0.002), the third auditory attention task (*p* < 0.001), and the fourth auditory attention task (*p* = 0.001).

Comparing nonpathological (NP) and pathological (P) results between the DMD and SLI groups showed significant difference for the following administered tests: word repetition (*p* = 0.047), nonword repetition (*p* = 0.003), sentence repetition (*p* = 0.022), speed (*p* = 0.001), and accuracy (*p* = 0.011) of information processing in the visual attention domain, in the first auditory attention task (*p* = 0.010), the second auditory attention task (*p* = 0.010), the third auditory attention task (*p* < 0.001), the fourth auditory attention task (*p* < 0.001), the first ST task (*p* < 0.001), and the second ST task (*p* < 0.001).

### 3.4. Cognitive Tests and Brain Dystrophin Involvement

Table 2 reports the results obtained in the four DMD subgroups according to the involvement of brain dystrophin isoforms. 

Of the 20 DMD boys, 9 had a mutation upstream of exon 44 and involvement of only Dp 427, 4 had mutations between 51 and 63 affecting both Dp427 and Dp140, and 2 patients had mutations after exon 63 with reduced expression of Dp427, Dp140, and Dp71. Five boys had mutations between 44 and 51 and were reported separately, as the involvement of Dp 140 in these patients is not always clear. 

Differences in performance according to the four brain dystrophin subgroups clinical variations were significant for the following administered tests: lexical complexity (*p* = 0.005) and word repetition (*p* = 0.002). The two children with abnormal results on the lexical complexity task were both in the DMD group with deletions above exon 62. Of the 4 children with abnormal results in the word repetition task, 2 had deletions above exon 51, and the other 2 had deletions above exon 62.

## 4. Discussion

A few studies reported that acquisition of early aspects of language appears to be impaired among young boys with DMD [10,11,12,16,17,18,22]. Parents of DMD boys report that they speak significantly later than both their unaffected siblings and their peers [22]. Speech delay is one of the possible presenting signs in young DMD boys [10,11,12], and lower scores on language subscales were reported in DMD boys with low global developmental quotients [16,17,18]. Our findings in preschool DMD boys confirm previous reports indicating language difficulties, and show that these can also be found in patients with normal DQ/IQ scores, and thereby not only as part of a more global delay.

Language difficulties appeared to be related to specific aspects of language development, while other aspects appeared to be more often spared. Lexical and syntactic comprehension, and some features of language production such as denomination appeared to be mainly spared, as, with two exceptions, the results were always within a normal range. Other aspects of language production such as articulation, in contrast, had abnormal results in 80% of the DMD patients. While the vocabulary was appropriate to the chronological age, with complete phonetic inventory, there were phonological simplifications, and in most of the DMD children (70%), several sounds were produced in an altered way. In our cohort, these difficulties were often found in the youngest children. Phonological difficulties were also present in 80% of DMD patients who had difficulties in performing more complex linguistic production, such as sentence repetition, while single-word production was generally well-preserved. There were also simplification processes that are characteristic of 3-year-old children, and atypical simplification processes. 

The observed profile in our DMD boys was different from that observed in the SLI group. While the SLI group had both phonetic and phonological deficits, DMD boys mainly had phonological problems.

The pattern of abnormal results is largely in agreement with the findings from other studies showing selective language involvement in older DMD boys [23,24], with sparing single-word comprehension and expression, and age-appropriate semantic verbal fluency [23,24]. As observed in our younger cases, most older DMD boys have limited phonological and morphosyntactic language processing [23,25,26,27,28,29,30], which was also not always related to a low IQ level [28]. A detailed analysis of the narratives also suggests the impairment of speech content [24], as the older boys used fewer verbs in their speech and had fewer complete sentences than what is expected of children their age. 

Different factors may contribute to the profile of language development observed in our data and in previous studies. As some findings could be compatible with attention deficits that were reported in DMD boys [3,14], we used additional tests to explore the possible role of these components. The examiners often reported that DMD boys had increasing difficulties with increasing testing time. When we used a battery of specifically developed tests for selective and sustained auditory attention, we found abnormal results in 90% of the DMD group with particular involvement on auditory tasks. On the test, they had greater difficulties in keeping their attention level stable for long periods (sustained auditory attention) than those in discriminating and selecting an auditory target from an interfering background (selective attention). These results suggest that difficulties in processing long sentences may be at least partly related to limited attentional resources necessary for processing, organizing, and correctly sequencing phonological information (the order of Ω sounds that compose the phonetic string). This is also suggested by the difficulties observed in tasks assessing working memory ability, such as nonword repetition test, especially with those that are long and contain consonant clusters, requiring a more complex phonological planning. 

Other possible contributing factors may be related to more general aspects of attention and hyperactivity. Although we did not systematically assess our patients using ADHD tests, as specific tests are not easily available in the preschool age range, clinical–behavioral observation in our cohort shows that the poor levels of attention and limited concentration capacity were associated with a tendency for patients to be easily distracted by objects around them, and with hyperkinetic behavior. This was also obvious from the results of the visual attention tests showing difficulties in accuracy and speed. 

Interestingly, LSD patients had fewer difficulties on visual attention rapidity, on auditory sustained attention, and more generally on attention. These differences support the hypothesis of possible different mechanisms underlying the language difficulties. 

The analysis of the involvement of the brain dystrophin isoforms related to the site of mutations in our young DMD children suggested that this may also play a role in some aspects of language difficulties. Some aspects such as articulation and sentence repetition were frequent in the whole cohort, but were relatively sparse in patients with preserved Dp140 and DP 71, while they were invariably affected in patients in whom these isoforms were involved. These data should be interpreted with caution, as they exclude patients with linguistic impairment as part of more global cognitive and developmental impairments that are more frequent in patients with the involvement of Dp140 and Dp71.

## 5. Conclusions

In conclusion, our results suggest that language difficulties can already be detected in young DMD boys with a pattern of difficulties that are consistent with what is reported in older DMD boys. Our results also suggest that these difficulties are possibly at least partly due to a sustained attention deficit, especially of the auditory channels, and more generally to reduced attention skills. These findings suggest that language abilities should be assessed in young DMD boys irrespective of their global neurodevelopmental quotient, as their early identification may facilitate rehabilitation plans that should include addressing the possible concomitant involvement of attentional skills and, when present, behavioral difficulties.

## Figures and Tables

**Figure 1 brainsci-12-01252-f001:**
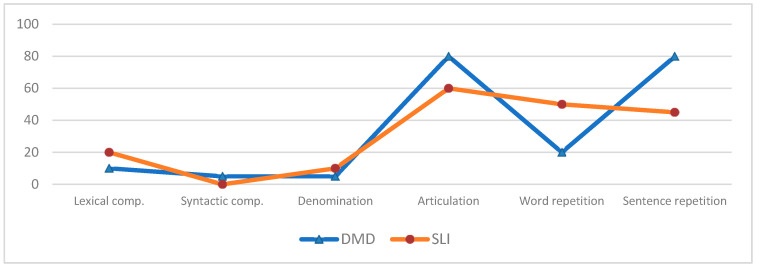
Percentage distribution of abnormal results on the language tests in the DMD and SLI groups.

**Figure 2 brainsci-12-01252-f002:**
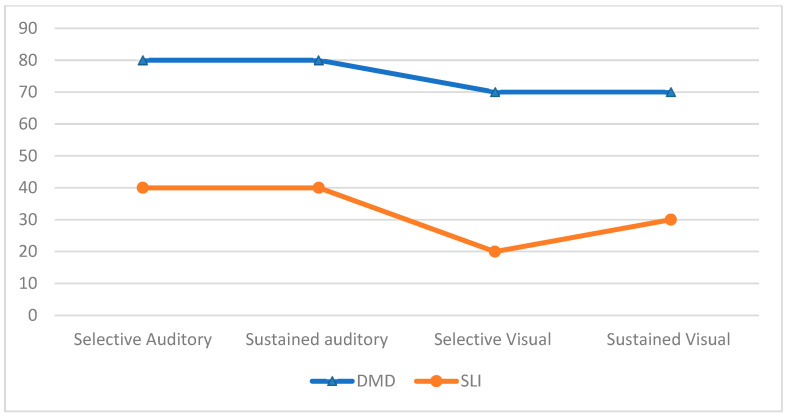
Percentage distribution of abnormal results on the attention tests in the DMD and SLI groups.

**Table 1 brainsci-12-01252-t001:** Comparison of the specific language impairment (SLI) and Duchenne muscular dystrophy (DMD) groups regarding administered tests.

Test	SLI (*n* = 20) N (%)	DMD (*n* = 20) N (%)	Mann–Whitney U Results	Chi-Squared/FET Results
*Lexical comprehension*			U = 175.000, z = −0.695, *p* = 0.512	*p* = 0.661, FET
NP	16 (80)	18 (90)		
P	4 (20)	2 (10)		
*Syntactic comprehension*			U = 206.000, z = 0.176, *p* = 0.883	
NP	20 (100)	20 (100)		
P	0 (0)	0 (0)		
*Denomination*				
NP	20 (100)	20 (100)	U = 174.500, z = −0.714, *p* = 0.495	
P	0 (0)	0 (0)		
*Articulation*			U = 209.500, z = 0.268, *p* = 0.799	*p* = 1.000, FET
NP	4 (20)	4 (20)		
P	16 (80)	16 (80)		
*Word repetition*			U = 112.500, z = –2.392, ***p* = 0.017 ***	X^2^ (1, N = 40) = 3.956, ***p* = 0.047 ***
NP	10 (50)	16 (80)		
P	10 (50)	4 (20)		
*Nonword repetition*			U = 312.000, z = 3.099, ***p* = 0.002 ***	***p*****= 0.003 ***, FET
NP	20 (100)	12 (60)		
P	0 (0)	8 (40)		
*Sentence repetition*			U = 272.000, z = 1.982, *p* = 0.052	X^2^ (1, N = 40) = 5.227, ***p* = 0.022 ***
NP	11 (55)	4 (20)		
P	9 (45)	16 (80)		
*Visual attention rapidity*			U = 320.500, z = 3.262, ***p* = 0.001 ***	X^2^ (1, N = 40) = 10.101, ***p* = 0.001 ***
NP	16 (80)	6 (30)		
P	4 (20)	14 (70)		
*Visual attention accuracy*			U = 258.500, z = 1.584, *p* = 0.114	X^2^ (1, N = 40) = 6.400, ***p* = 0.011 ***
NP	14 (70)	6 (30)		
P	6 (30)	14 (70)		
*Auditory attention (ACPT1)*			U = 263.500, z = 1.729, *p* = 0.086	X^2^ (1, N = 40) = 6.667, ***p* = 0.010 ***
NP	12 (60)	4 (20)		
P	8 (40)	16 (80)		
*Auditory attention (ACPT2)*			U = 313.500, z = 3.078, ***p* = 0.002 ***	X^2^ (1, N = 40) = 6.667, ***p* = 0.010 ***
NP	12 (60)	4 (20)		
P	8 (40)	16 (80)		
*Auditory attention (ACPT3)*			U = 375.000, z = 4.747, ***p* < 0.001 ***	X^2^ (1, N = 40) = 15.000, ***p* < 0.001 ***
NP	14 (70)	2 (10)		
P	6 (30)	18 (90)		
*Auditory attention (ACPT4)*			U = 316.000, z = 3.164, ***p* = 0.001 ***	X^2^ (1, N = 40) = 10.989, ***p* < 0.001 ***
NP	12 (60)	2 (10)		
P	8 (40)	18 (90)		
*ST1*			U = 272.000, z = 2.001, *p* = 0.052	X^2^ (1, N = 40) = 10.989, ***p* < 0.001 ***
NP	12 (60)	2 (10)		
P	8 (40)	18 (90)		
*ST2*			U = 270.000, z = 1.936, *p* = 0.060	X^2^ (1, N = 40) = 10.989, ***p* < 0.001 ***
NP	12 (60)	2 (10)		
P	8 (40)	18 (90)		

Abbreviations: NP, not pathological; P, pathological. * significance at *p* < 0.05.

**Table 2 brainsci-12-01252-t002:** Comparison within Duchenne muscular dystrophy (DMD) group regarding administered tests.

Test	DMD 44 (*n* = 9) N (%)	DMD 45–51 (*n* = 5) N (%)	DMD > 51 (*n* = 4) N (%)	DMD > 62 (*n* = 2) N (%)	FET Results
*Lexical comprehension*					***p* = 0.005 ***
NP	9 (100)	5 (100)	4 (100)	0 (100)	
P	0 (0)	0 (0)	0 (0)	2 (100)	
*Syntactic comprehension*					
NP	9 (100)	5 (100)	4 (100)	2 (100)	
P	0 (0)	0 (0)	0 (0)	0 (0)	
*Denomination*					
NP	9 (100)	5 (100)	4 (100)	2 (100)	
P	0 (0)	0 (0)	0 (0)	0 (0)	
*Articulation*					*p* = 0.167
NP	4 (44.4)	0 (0)	0 (0)	0 (100)	
P	5 (55.6)	5 (100)	4 (100)	2 (100)	
*Word repetition*					***p* = 0.002 ***
NP	9 (100)	5 (100)	2 (50)	0 (100)	
P	0 (0)	0 (0)	2 (50)	2 (100)	
*Nonword repetition*					*p* = 0.239
NP	7 (77.8)	3 (60)	2 (50)	0 (100)	
P	2 (22.2)	2 (40)	2 (50)	2 (100)	
*Sentence repetition*					*p* = 0.167
NP	4 (44.4)	0 (0)	0 (0)	0 (100)	
P	5 (55.6)	5 (100)	4 (100)	2 (100)	
*Visual attention rapidity*					*p* = 0.430
NP	4 (44.4)	2 (40)	0 (0)	0 (100)	
P	5 (55.6)	3 (60)	4 (100)	2 (100)	
*Visual attention accuracy*					*p* = 0.430
NP	4 (44.4)	2 (40)	0 (0)	0 (100)	
P	5 (55.6)	3 (60)	4 (100)	2 (100)	
*Auditory attention (ACPT1)*					*p* = 0.167
NP	4 (44.4)	0 (0)	0 (0)	0 (100)	
P	5 (55.6)	5 (100)	4 (100)	2 (100)	
*Auditory attention (ACPT2)*					*p* = 0.167
NP	4 (44.4)	0 (0)	0 (0)	0 (100)	
P	5 (55.6)	5 (100)	4 (100)	2 (100)	
*Auditory attention (ACPT3)*					*p* = 0.763
NP	2 (22.2)	0 (0)	0 (0)	0 (100)	
P	7 (77.8)	5 (100)	4 (100)	2 (100)	
*Auditory attention (ACPT4)*					*p* = 0.763
NP	2 (22.2)	0 (0)	0 (0)	0 (100)	
P	7 (77.8)	5 (100)	4 (100)	2 (100)	
*ST1*					*p* = 0.763
NP	2 (22.2)	0 (0)	0 (0)	0 (100)	
P	7 (77.8)	5 (100)	4 (100)	2 (100)	
*ST2*					*p* = 0.763
NP	2 (22.2)	0 (0)	0 (0)	0 (100)	
P	7 (77.8)	5 (100)	4 (100)	4 (100)	

Abbreviations: NP, not pathological; P, pathological; DMD, Duchenne muscular dystrophy upstream 44; DMD 45–51, Duchenne muscular dystrophy exon 45–51; DMD > 51, Duchenne muscular dystrophy exon > 51; DMD > 62, Duchenne muscular dystrophy exon > 62. * Shows significance at *p* < 0.05.

## Data Availability

The data that support the findings of this study are available on request from the corresponding author. The data are not publicly available due to privacy or ethical restrictions.

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
