# Peer review of "Language Development in Preschool Duchenne Muscular Dystrophy Boys"

_brainsci, 2022, doi:10.3390/brainsci12091252_

Round 1

Reviewer 1 Report

Dear Authors,

I read your very interesting work and here i enclose my recommendations to you:

Strengths: The Introduction and the Methods section are very well documented and the results according to the analysis that was performed are good. 

Weaknesses: The statistics section of this work needs an update I believe. I suggest the authors besides chi-square tests to provide comparisons of means of the assessments that were used. The typical TD peers to the DMD sample  this could be easy to be addressed. Additionally, with the comparison of those two subgroups we can see only difference in language development fo children with DMD and not their track in conquering language milestones.

Furthermore, the Authors discuss the language development of children with DMD it is expected to see follow ups and the development of language and phonology for example in 3, 6, 9 and 12 months. 

The discussion has a rational behind bust since statistical analysis and results are missing according to the above comment, I suggest the Authors to consider this and re-write this section. 

Thank you.

Author Response

Dear colleague, thank you for contribution in ameliorating our study. We will provide comparisons of means of the assessments that were used and discuss it. We will re-write the section as per your suggestion. We will also think about future follow ups to collect data on the development of language and phonology along the months.

Reviewer 2 Report

The authors have studied the language difficulties in DMD boys. The authors claimed that the defect in language learning in DMD boys is due to attention deficit. The study is mostly complete. However, there are several demerits in the study that needs to be addressed. My comments are provided below

1.       The study is only focused on DMD boys and did not include all the gender.

2.       The sample size is very low in Table 2. For example, the sample size in the last column is two. The authors should provide a justification for it.

3.     The muscle weakness in DMD patients is an important factor. These include muscle from hearing and auditory system which may affect the learning process. The authors should discuss this point.

4.        The variation in dialect of the language (mother tongue) may also affect the language learning ability of the DMD patients. The authors did not discuss about it.

5.       The authors did not include the data from low IQ non DMD population.

Author Response

Dear colleague, thank you for contribution in ameliorating our study.

The study as you highlighted is only focused on DMD boys and did not include all the gender primarily because DMD usually affects males. Surely it could be interesting to develop future research in order to study also female population.  

We will add some informations about the muscle weakness in DMD patients as an important factor which may affect the learning process.
The variation in dialect of the language (mother tongue) may also affect the language learning ability of the DMD patients. Let’s say in Italy, in spite of differences from one region to another, national educational programs for primary school tend to level the differences in the Italian mother-tongue population (all the boys assessed were Italian mothertongue). It could be in any case interesting to develop in the future this point in a dedicated study investigating differences from one Italian district to another in language learning on a most huge sample.

Round 2

Reviewer 1 Report

Dear Authors,

I read your updated manuscript. I am happy that you have addressed all my suggestions. The last suggestions is to have a minor check for spelling, typing and syntax errors. 

Thank you.

Reviewer 2 Report

The authors have addressed all my concerns in the revised manuscript. I support the publication of the manuscript.